# Relative entropic uncertainty relation for scalar quantum fields

**Stefan Floerchinger**[*], **Tobias Haas**[†] **and Markus Schröfl**[‡]

Institut für Theoretische Physik, Universität Heidelberg,
Philosophenweg 16, 69120 Heidelberg, Germany

[*] stefan.floerchinger@thphys.uni-heidelberg.de, [†] t.haas@thphys.uni-heidelberg.de
[‡] m.schroefl@thphys.uni-heidelberg.de

## Abstract

Entropic uncertainty is a well-known concept to formulate uncertainty relations for continuous variable quantum systems with finitely many degrees of freedom. Typically, the bounds of such relations scale with the number of oscillator modes, preventing a straightforward generalization to quantum field theories. In this work, we overcome this difficulty by introducing the notion of a functional relative entropy and show that it has a meaningful field theory limit. We present the first entropic uncertainty relation for a scalar quantum field theory and exemplify its behavior by considering few particle excitations and the thermal state. Also, we show that the relation implies the multidimensional Heisenberg uncertainty relation.

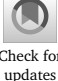

## Contents



# 1   Introduction

The uncertainty principle is one of the most well-known features of quantum mechanics. It is a consequence of the non-commuting nature of certain quantum observables and makes a statement about how precisely two observables $A$ and $B$ can be prepared or measured at a certain instance of time [1–3].

If we explicitly consider the case of the continuous variables position $X$ and momentum $P$ of a single mode, which fulfill the usual commutation relation $[X,P] = i$, the famous variance-based uncertainty relation by Heisenberg and Kennard reads [1,2]

$$\sigma_x^2 \sigma_p^2 \geq \frac{1}{4}. \tag{1}$$

During the last decades, uncertainty relations have mainly been studied from a quantum information theoretic perspective, emerging in the wide field of entropic uncertainty relations (see [4–6] for reviews). The entropic analog to Eq. (1) was formulated by Białynicki-Birula and Mycielski for the probability densities $f(x)$ and $g(p)$ [7–10]

$$S(f) + S(g) \geq 1 + \ln \pi, \tag{2}$$

where $S(f) = -\int \mathrm{d}x\, f(x) \ln f(x)$ is the differential entropy.

Nowadays, many entropic uncertainty relations have been proven and studied, for example the Maassen-Uffink entropic uncertainty relation for observables with discrete spectrum formulated in terms of Shannon entropies [11–14], the information exclusion principles in terms of mutual information [15–17], for Rényi entropies [13,18], for Wehrl entropies [19,20], in terms of conditional entropies in the presence of (quantum) memory [14,21–24], to quantify uncertainty between energy and time [25] or in a more general setting of complementary operator algebras [26–28]. Furthermore, the two different cases of discrete and continuous variables have been unified in [29,30].

In this work, we extend the concept of entropic uncertainty to scalar quantum field theories, for which our motivation is threefold. First, the information theoretic point of view has lead to many insights into quantum field theories, most prominently in the contexts of entanglement [31–33], thermalization [34–36] and black hole physics [37–39]. As the uncertainty principle is central to every quantum theory of nature, a rigorous entropic formulation for quantum fields is essential for a deeper understanding of quantum field theories.

Second, uncertainty relations play an important role for witnessing entanglement, especially for continuous variable quantum systems. Besides the prominent second-order inseparability criteria by Simon [40] and Duan et al. [41], there exist stronger entropic criteria [42–44] based on entropic uncertainty relations. Also, entropic uncertainty relations can be used to formulate steering inequalities [45,46], or, by including (quantum) memory [24], one can derive bounds on entanglement measures [47]. For experimental applications of entropic criteria see e.g. [45,47].

When trying to extend notions of entanglement measures from the case of finite number of modes to field theories one encounters the problem that these measures diverge even for the vacuum state [48]. Also, formulating criteria to certify entanglement for quantum fields requires a different line of reasoning compared to their finite dimensional analogs. Therefore, a well-defined notion of entropic uncertainty for quantum fields paves the ground for field-theoretic entropic entanglement witnesses, which could potentially be applicable to a large class of experimental setups.

Third, the transition from quantum mechanics to quantum field theory is known to be non-trivial. Several divergencies of different origins arise when taking the continuum and infinite volume limits, requiring a careful analysis of all quantities. Hence, the extension of entropic uncertainty for a finite number of modes to continuous fields can be considered an interesting task by itself.

To that end, we propose the use of relative entropy, which keeps many of its useful properties also in limiting cases. In particular, when considering the relative entropy between a discrete distribution $p(x)$ and some reference $\tilde{p}(x)$, which is defined as [49]

$$S(p\|\tilde{p}) = \sum_x p(x)(\ln p(x) - \ln \tilde{p}(x)),\tag{3}$$

one can take the continuum limit $p(x) \to f(x)\mathrm{d}x, \tilde{p}(x) \to \tilde{f}(x)\mathrm{d}x$, where $f(x)$ and $\tilde{f}(x)$ denote probability density functions, to obtain the differential relative entropy

$$S(f\|\tilde{f}) = \int_x \mathrm{d}x\, f(x)\big(\ln f(x) - \ln \tilde{f}(x)\big).\tag{4}$$

A similar line of reasoning fails for the Shannon entropy $S(p)$, in the sense that the differential entropy $S(f)$ can only be obtained from $S(p)$ when adding an infinite constant [50]. As a consequence, the differential entropy $S(f)$ is not non-negative, as opposed to the Shannon entropy $S(p)$. In contrast, the relative entropies $S(p\|\tilde{p})$ and $S(f\|\tilde{f})$ are both non-negative and zero if and only of the two arguments agree. Hence, the notion of *distinguishability* measured in terms of relative entropy may be considered to be more universal than the notion of *missing information* measured in terms of entropy.

Motivated by these properties of the relative entropy, we have unified discrete and continuous entropic uncertainty relations in Ref. [51], leading to an upper bound for a sum of two relative entropies with respect to maximum entropy model distributions. In this work, we extend this idea to free scalar quantum fields, to obtain the relative entropic uncertainty relation given in eq. (52), which holds for a collection of oscillators as well as for (possibly averaged) scalar quantum fields. While many traditional (entropic) uncertainty relations are lower bounds on sums of entropies (cf. e.g. (1) and (2)), the relative entropic uncertainty relation (52) provides a non-trivial upper bound for a sum of two relative entropies in terms of differences of two-point correlation functions. In this relation the uncertainty principle is encoded in the sense that this upper bound is finite, similar to the finite lower bound in (2).

We will develop the idea of a field-theoretic relative entropic uncertainty relation by means of a free scalar field theory. In this case, the field operator and the conjugate momentum field operator fulfill a bosonic commutation relation (analogous to position and momentum in quantum mechanics) and the vacuum state has a Gaussian Schrödinger functional. Our relation (52) holds for all states within this setup and one can expect that the construction can be generalized to interacting scalar theories at least in the perturbative regime, with the caveat that the bound may acquire additional terms containing higher-order correlation functions. For field theories constrained by other algebras, for example non-abelian gauge theories, the bound may differ substantially.

**The remainder of the paper is organized as follows.**    In section 2 we introduce the Schrödinger functional formalism together with functional probability distributions of selected states, such as coherent, excited and thermal states. Also, we briefly mention the multidimensional Heisenberg relation in quantum field theory. Then, we show that the notion of a functional entropy is ill-defined, which serves as a motivation for the functional relative entropy as a meaningful measure for entropic uncertainty with respect to coherent states in section 3. We derive our relative entropic uncertainty relation and demonstrate its independence of the number of oscillator modes by considering examples, namely excited states and the thermal state, also in the field theory limit. Furthermore, we show that the relative entropic uncertainty relation implies the Heisenberg relation and discuss in which sense our findings are accessible in experiments. Finally, we summarize our findings and give an outlook in section 4.

**Notation.**    Throughout this paper we employ natural units $\hbar = c = k_{\mathrm{B}} = 1$ and disregard operator hats. Instead, we use capital letters for operators and small letters for their eigenvalues and eigenvectors, the only exception to this rule being creation and annihilation operators $a^\dagger, a$, respectively. As a consequence, we also denote random variables by small letters. Also, we refer to vacuum quantities by using a bar, e.g., $\bar{F}[\phi]$.

## 2  Functional probability densities

We start with an introductory section, where we set up the discrete as well as the continuous theory, define the notion of functional probability densities within the Schrödinger functional formalism and introduce the multidimensional Heisenberg uncertainty relation.

### 2.1  From oscillator modes to a quantum field theory

To keep the connection to quantum mechanics rather close, let us begin with a collection of coupled harmonic oscillators on a one-dimensional spatial lattice. This approach has the advantage that the dependence on the number of modes $N$ of the bound of an entropic uncertainty relation becomes manifest. We choose a one-dimensional model out of convenience, and it should be noted that the following discussion can easily be generalized to an arbitrary number of spatial dimensions.

The Hamiltonian of our interest reads

$$H = \frac{1}{2}\sum_j \varepsilon \left[ \pi_j^2 + \frac{1}{\varepsilon^2}\left(\phi_j - \phi_{j-1}\right)^2 + m^2\phi_j^2 \right], \tag{5}$$

where $j \in \{1,...,N\}$ labels spatial positions with $N$ being an integer. Therein, the displacement from the equilibrium position of the $j$th oscillator is denoted by the real field $\phi_j$ (the corresponding conjugate momentum is $\pi_j$) and we assume periodic boundary conditions, i.e., $\phi_N = \phi_0$. Furthermore, $\varepsilon$ is the lattice constant, such that $1/\varepsilon$ provides an ultra-violet regulator. The lattice constant has been introduced as a precursor for the continuum limit $\varepsilon \to 0$. In contrast, the oscillator picture can be recovered by setting $\varepsilon = 1$.

The Hamiltonian (5) can be diagonalized by performing a discrete Fourier transform

$$\phi_j = \sum_\ell \frac{\Delta k}{2\pi} e^{i\Delta k\ell\varepsilon j}\tilde{\phi}_\ell, \quad \pi_j = \sum_\ell \frac{\Delta k}{2\pi} e^{-i\Delta k\ell\varepsilon j}\tilde{\pi}_\ell, \tag{6}$$

where the integer valued index $-\frac{N}{2} \le \ell < \frac{N}{2}$ labels the momentum modes. The length of the system is $L = N\varepsilon$ and the lattice spacing in $k$-space is $\Delta k = \frac{2\pi}{L}$. Note also the relations $\tilde{\phi}_\ell^* = \tilde{\phi}_{-\ell}$ and $\tilde{\pi}_\ell^* = \tilde{\pi}_{-\ell}$.

At this point, we note that the coordinates $\tilde{\phi}$ and $\tilde{\pi}$ are complex quantities. As we wish to work only with real valued configurations, we employ another unitary transformation

$$\tilde{\phi}_\ell = \frac{1}{2}(1+i)\phi_\ell + \frac{1}{2}(1-i)\phi_{-\ell},$$
$$\tilde{\pi}_\ell = \frac{1}{2}(1-i)\pi_\ell + \frac{1}{2}(1+i)\pi_{-\ell}, \tag{7}$$

with $\phi_\ell, \pi_\ell \in \mathbb{R}$. In terms of these quantities, the Hamiltonian reads

$$H = \frac{1}{2}\sum_\ell \frac{\Delta k}{2\pi}\left[\pi_\ell^2 + \omega_\ell^2 \phi_\ell^2\right], \tag{8}$$

with frequencies

$$\omega_\ell \equiv \sqrt{\frac{4}{\varepsilon^2}\sin^2\left(\frac{\Delta k \ell \varepsilon}{2}\right) + m^2}. \tag{9}$$

Eq. (8) corresponds now to a Hamiltonian of decoupled modes. To emphasize the oscillator picture, one can set $L = 1$, which implies $\Delta k = 2\pi$.

The quantization procedure is straightforward. We impose canonical commutation relations on the hermitian quantum field operators in momentum space

$$[\Phi_\ell, \Pi_{\ell'}] = i\frac{2\pi}{\Delta k}\delta_{\ell\ell'}, \quad [\Phi_\ell, \Phi_{\ell'}] = [\Pi_\ell, \Pi_{\ell'}] = 0. \tag{10}$$

Then, we define creation and annihilation operators as

$$a_\ell \equiv \frac{1}{\sqrt{2\omega_\ell}}(\omega_\ell \Phi_\ell + i\Pi_\ell),$$
$$a_\ell^\dagger \equiv \frac{1}{\sqrt{2\omega_\ell}}(\omega_\ell \Phi_\ell - i\Pi_\ell), \tag{11}$$

which satisfy $[a_\ell, a_{\ell'}^\dagger] = \frac{2\pi}{\Delta k}\delta_{\ell\ell'}$. In terms of these ladder operators, the Hamilton operator takes the form

$$H = \sum_\ell \frac{\Delta k}{2\pi}\omega_\ell\left(a_\ell^\dagger a_\ell + \frac{1}{2}\frac{2\pi}{\Delta k}\right), \tag{12}$$

where the second term contains the well-known divergence of the vacuum energy in the field theory limit.

The chain of $N$ coupled harmonic oscillators shall, in the following discussion, serve as a model which allows us to make any dependencies on the number of modes $N$ explicit. By taking the continuum limit of the discrete theory defined in (5), i.e., taking $\varepsilon \to 0$, $N \to \infty$ and keeping $L = N\varepsilon$ fixed, leading to $\phi_j \equiv \phi(\varepsilon j) \to \phi(x)$ with $x \in [0, L]$ (similar for the conjugate field $\pi$), we obtain a relativistic quantum field theory for a free massive scalar field with periodic boundary conditions. In this case, the momenta are still discrete, but are drawn from an unbounded set, i.e. $\ell \in \mathbb{Z}$.

The infinite volume limit is obtained by taking $L \to \infty$ (or equivalently $\Delta k \to 0$) and $N \to \infty$, with $\varepsilon = \frac{L}{N}$ fixed. In this limit, $\phi_\ell \equiv \phi(\Delta k \ell) \to \phi(p)$ with $p \in [-\frac{\pi}{\varepsilon}, \frac{\pi}{\varepsilon}]$ and the commutation relations in (10) have to be understood under an integral with integral measure $\frac{dp}{2\pi}$, i.e.,

$$[\Phi(p), \Pi(q)] = i\delta(p-q),$$
$$[\Phi(p), \Phi(q)] = [\Pi(p), \Pi(q)] = 0. \tag{13}$$

The field theory limit corresponds to taking both the continuum as well as the infinite volume limit, such that the discrete Fourier transform in (6) becomes

$$\phi(x) = \int \frac{\mathrm{d}p}{2\pi} e^{ipx} \tilde{\phi}(p), \quad \pi(x) = \int \frac{\mathrm{d}p}{2\pi} e^{-ipx} \tilde{\pi}(p). \tag{14}$$

In this case, we obtain a relativistic quantum field theory on an infinite interval with Hamiltonian

$$H = \frac{1}{2} \int \mathrm{d}x \left[ \pi^2(x) + (\partial_x \phi(x))^2 + m^2 \phi^2(x) \right], \tag{15}$$

and relativistic dispersion relation $\omega(p) = \sqrt{p^2 + m^2}$.

## 2.2 Schrödinger functional formalism

Often, quantum field theory is treated within the Heisenberg picture, i.e., time-dependent observables but time-independent states. In this work, we employ the Schrödinger picture, such that observables are time-independent while states depend on time. More precisely, the state represented by a density operator $\rho$ is defined on a Cauchy hypersurface $\Sigma_t$, for example at an instance of time $t$.

It is convenient to introduce complete orthonormal sets of eigenstates at this time $t$, which are the eigenstates of the field operator,

$$\Phi_\ell |\phi\rangle = \phi_\ell |\phi\rangle, \tag{16}$$

as well as of the conjugate momentum field operator

$$\Pi_m |\pi\rangle = \pi_m |\pi\rangle, \tag{17}$$

where $\phi_\ell$ and $\pi_m$ represent the corresponding eigenvalues, which are real-valued (cf. (7)).

Throughout the remainder of this work, the indices $\ell$ and $m$ denote momentum modes and have to be understood in terms of a regularized theory as discussed in subsection 2.1. At some particular points we may wish to emphasize certain properties of the continuum (or additionally the infinite volume) limit explicitly, in which case we employ the usual continuum notation for momenta, i.e. $p$ and $q$. However, for most expressions (especially for bilinear forms) the field theory limit can be employed straightforwardly.

In the Schrödinger picture[1], the momentum operator can be represented as a functional derivative

$$\Pi_m = -i \frac{\delta}{\delta \phi_m}, \tag{18}$$

and the matrix representation of the density operator $\rho$ reads [35, 52]

$$\rho[\phi_+, \phi_-] = \langle \phi_+ | \rho | \phi_- \rangle. \tag{19}$$

The density of this matrix defines the functional probability distribution

$$F[\phi] = \rho[\phi, \phi] = \langle \phi | \rho | \phi \rangle, \tag{20}$$

which formally corresponds to the probability density of finding the quantum field $\Phi_\ell$ in the configuration $\phi_\ell$ if measuring in the field basis $|\phi\rangle$ at time $t$.

The functional probability distribution is non-negative $F[\phi] \geq 0$ and normalized to unity $\int \mathcal{D}\phi \, F[\phi] = \mathrm{Tr}\{\rho\} = 1$. For any pure state $\rho = |\psi\rangle\langle\psi|$ the entries of the density matrix in

---

[1]Without loss of generality we consider the field $\phi$ in the following, but all steps can be repeated for the conjugate field $\pi$.

the eigenfield basis reduce to a scalar product of ordinary Schrödinger wave functionals, which are defined as $\Psi[\phi] = \langle \phi | \psi \rangle$. Thus, we obtain Born's probability rule [53] for the probability density functional of a quantum field

$$F[\phi] = |\Psi[\phi]|^2, \tag{21}$$

justifying the physical intuition stated above.

In the Schrödinger functional formalism, expectation values can be computed from functional integrals

$$
\begin{aligned}
\langle \phi_\ell \rangle &= \int \mathcal{D}\phi \, \phi_\ell \, F[\phi], \\
\langle \pi_m \rangle &= -i \int \mathcal{D}\phi \, \frac{\delta \rho[\phi_+, \phi_-]}{\delta \phi_{+m}} \bigg|_{\phi_+ = \phi_- = \phi}.
\end{aligned}
\tag{22}
$$

Therein, the functional integral measure is given by

$$\int \mathcal{D}\phi = \prod_\ell \int \mathrm{d}\phi_\ell \sqrt{\frac{\Delta k}{2\pi}}. \tag{23}$$

The more general $n$-point correlation functions can be computed analogously. For the special case of $n = 2$, we denote the connected two-point correlators by

$$
\begin{aligned}
\mathcal{M}_{\ell m} &= \langle \phi_\ell \phi_m \rangle - \langle \phi_\ell \rangle \langle \phi_m \rangle, \\
\mathcal{N}_{\ell m} &= \langle \pi_\ell \pi_m \rangle - \langle \pi_\ell \rangle \langle \pi_m \rangle,
\end{aligned}
\tag{24}
$$

which will play a crucial role for the relative entropic uncertainty relation.

What we have described so far is known as a homodyne measurement, that is measuring the state $\rho$ in the field and momentum field eigenbases. There also exists the possibility of a heterodyne measurement, as described by the Husimi $Q$-functional, a joint functional probability density in the field theory phase space. It is defined as the measured distribution when employing a positive operator-valued measure with respect to pure coherent state projectors $\rho = |\alpha\rangle \langle \alpha|$, with $\alpha = (\phi + i\pi)/\sqrt{2}$,

$$Q[\phi, \pi] = \mathrm{Tr}\{\rho \, |\alpha\rangle \langle \alpha|\} = \langle \alpha | \rho | \alpha \rangle. \tag{25}$$

We will report elsewhere on the formulation of a relative entropic uncertainty relation associated with the latter functional density and proceed here with the marginal functional density (20) and a similarly defined object for conjugate momenta $\pi_m$.

## 2.3 Functional probability densities for selected states

Let us now introduce the functional probability densities for those states to which we will apply our relative entropic uncertainty relation in section 3. We start with the vacuum and coherent states of the scalar quantum field theory, proceed with constructing excited states and finish with a discussion of the thermal state.

### 2.3.1 Vacuum and coherent states

The vacuum wave functional $\bar{\Psi}[\phi]$ can be obtained by solving the stationary Schrödinger equation $H\bar{\Psi}[\phi] = \bar{E}\bar{\Psi}[\phi]$ with the Hamiltonian $H$ given in Eq. (8) and by using the functional

derivative representation of the momentum operator (18) (analogously for the momentum field $\pi$), leading to

$$\bar{\Psi}[\phi] = \frac{1}{\sqrt{\bar{Z}_\phi}} \exp\left(-\frac{1}{4} \sum_\ell \frac{\Delta k}{2\pi} \sum_m \frac{\Delta k}{2\pi} \phi_\ell \, \bar{\mathcal{M}}_{\ell m}^{-1} \, \phi_m \right),$$

$$\bar{\Psi}[\pi] = \frac{1}{\sqrt{\bar{Z}_\pi}} \exp\left(-\frac{1}{4} \sum_\ell \frac{\Delta k}{2\pi} \sum_m \frac{\Delta k}{2\pi} \pi_\ell \, \bar{\mathcal{N}}_{\ell m}^{-1} \, \pi_m \right),$$

(26)

with normalization constants

$$\bar{Z}_\phi = \prod_\ell \sqrt{\frac{\pi}{\omega_\ell}}, \quad \bar{Z}_\pi = \prod_\ell \sqrt{\pi \omega_\ell}, \tag{27}$$

and, for a discrete set of modes,

$$\bar{\mathcal{M}}_{\ell m}^{-1} = \frac{2\pi}{\Delta k} 2\omega_\ell \delta_{\ell m}, \quad \bar{\mathcal{N}}_{\ell m}^{-1} = \frac{2\pi}{\Delta k} \frac{2}{\omega_\ell} \delta_{\ell m}. \tag{28}$$

Inverting the latter matrices according to

$$\sum_m \frac{\Delta k}{2\pi} \bar{\mathcal{M}}_{\ell m}^{-1} \bar{\mathcal{M}}_{mn} = \frac{2\pi}{\Delta k} \delta_{\ell n}, \tag{29}$$

leads to discrete covariance matrices

$$\bar{\mathcal{M}}_{\ell m} = \frac{2\pi}{\Delta k} \frac{1}{2\omega_\ell} \delta_{\ell m}, \quad \bar{\mathcal{N}}_{\ell m} = \frac{2\pi}{\Delta k} \frac{\omega_\ell}{2} \delta_{\ell m}. \tag{30}$$

In the field theory limit we instead obtain expressions which have to be understood in a distributional sense, i.e.,

$$\int \frac{\mathrm{d}q}{2\pi} \bar{\mathcal{M}}^{-1}(p, q) \bar{\mathcal{M}}(q, k) = 2\pi \delta(p - k), \tag{31}$$

and

$$\bar{\mathcal{M}}(p, q) = \frac{\pi}{\omega(p)} \delta(p - q), \ \bar{\mathcal{N}}(p, q) = \pi \omega(p) \delta(p - q). \tag{32}$$

Note also that in this case the formally infinite normalization constants $\bar{Z}_\phi$ and $\bar{Z}_\pi$ decompose into an infinite product over discrete vacuum contributions.

Compairing eqs. (30) and (32) reveals the difficulties of the field theory limit. While the discrete covariance matrices have well-defined diagonal elements, the distributional character of the continuous covariance matrices implies that formally $\bar{\mathcal{M}}(p, p) \sim \delta(0)$. Therefore, covariance matrices have to be understood under an integral in the field theory limit.

Also, the two Schrödinger wave functionals in eq. (26) are of the same form, which turns out to be a generic feature. Therefore, we only state expressions for the field $\phi_\ell$ in the following, keeping in mind that the corresponding expressions for the momentum field $\pi_m$ can be obtained analogously when replacing the covariance matrix accordingly.

The vacuum functional probability density can be obtained using Born's rule (21) and reads

$$\bar{F}[\phi] = \frac{1}{\bar{Z}_\phi} \exp\left(-\frac{1}{2} \sum_\ell \frac{\Delta k}{2\pi} \sum_m \frac{\Delta k}{2\pi} \phi_\ell \, \bar{\mathcal{M}}_{\ell m}^{-1} \phi_m \right). \tag{33}$$

The vacuum is a special case of a coherent state, a class of states which minimize all uncertainty relations and are therefore of special interested to us (see section 3). Coherent states follow

from displacing the vacuum state in phase space. Consequently, they are parameterized by a complex field $\alpha_\ell$, which is given by

$$\alpha_\ell = \frac{1}{\sqrt{2}} \left( \phi_\ell^\alpha + i \pi_\ell^\alpha \right), \tag{34}$$

where we used the notation $\langle \phi_\ell \rangle_\alpha \equiv \phi_\ell^\alpha$ and $\langle \pi_\ell \rangle_\alpha \equiv \pi_\ell^\alpha$, such that the vacuum corresponds to $\phi_\ell^\alpha = \pi_\ell^\alpha = 0$. Then, the coherent wave functional reads

$$F_\alpha[\phi] = \frac{1}{\bar{Z}_\phi} \exp\left( -\frac{1}{2} \sum_\ell \frac{\Delta k}{2\pi} \sum_m \frac{\Delta k}{2\pi} (\phi_\ell - \phi_\ell^\alpha) \bar{\mathcal{M}}_{\ell m}^{-1} (\phi_m - \phi_m^\alpha) \right), \tag{35}$$

where the covariances agree with the vacuum covariance.

### 2.3.2 Excited states

An interesting class of states are excited states as they typically allow for a (quasi-)particle interpretation, also in the field theory limit. An excited state with $n_k$ excitations in mode $k$ can be obtained by acting with $n_k$ creation operators $a_k^\dagger$ defined in (11) on the vacuum (see e.g. [52]). In the Schrödinger picture, the creation and annihilation operators can be represented in terms of a functional derivative using (18),

$$\begin{aligned} a_\ell &= \frac{1}{\sqrt{2\omega_\ell}} \left( \omega_\ell \phi_\ell + \frac{\delta}{\delta \phi_\ell} \right), \\ a_\ell^\dagger &= \frac{1}{\sqrt{2\omega_\ell}} \left( \omega_\ell \phi_\ell - \frac{\delta}{\delta \phi_\ell} \right). \end{aligned} \tag{36}$$

To allow for several modes being excited simultaneously, possibly also multiple times, we introduce an index set $\mathfrak{I}$, such that every mode $k \in \mathfrak{I}$ carries $n_k$ excitations, while all other modes remain in their respective ground states. Then, the wave functional $\Psi[\phi]$ of such a state is

$$\Psi[\phi] = \prod_{k \in \mathfrak{I}} \frac{1}{\sqrt{n_k!}} \left( \sqrt{\frac{\Delta k}{2\pi}} a_k^\dagger \right)^{n_k} \bar{\Psi}[\phi], \tag{37}$$

where the involved factors guarantee the correct normalization. One can show that the latter wave functional can be rewritten using the probabilist's Hermite polynomials. Then, the corresponding functional probability density reads

$$F[\phi] = \prod_{k \in \mathfrak{I}} \frac{1}{n_k!} H_{n_k}^2 \left( \frac{\phi_k}{\sqrt{\bar{\mathcal{M}}_{kk}}} \right) \bar{F}[\phi], \tag{38}$$

wherein the probabilist's Hermite polynomials $H_{n_k}(\phi_k)$ are given by

$$H_{n_k}(\phi_k) = n_k! \sum_{\gamma=0}^{\lfloor \frac{n_k}{2} \rfloor} \frac{(-1)^\gamma \phi_k^{n_k - 2\gamma}}{\gamma! (n_k - 2\gamma)! \, 2^\gamma}. \tag{39}$$

It should be noted that normalizing excited states in the field theory limit requires additional formally infinite factors compared to the vacuum state, which is again a consequence of $\bar{\mathcal{M}}(k, k) \sim \delta(0)$. Mathematically, this is due to the fact that $\Phi(p)$ and $\Pi(p)$, respectively, are not operators but rather operator-valued distributions. Some implications of this property are discussed in subsection 3.7. However, we will later show that this will not introduce additional problems when it comes to relative entropic uncertainty.

### 2.3.3 Thermal state

Another interesting example is the thermal state. It follows from the maximum entropy principle for a fixed energy expectation value [54–57] or equivalently from the recently introduced principle of minimum expected relative entropy [58]. It reads

$$\rho_T = \frac{1}{Z} e^{-\beta H}, \tag{40}$$

where $\beta = 1/T$ denotes the inverse temperature and $Z$ is the canonical partition function.

The resulting functional distribution is again of Gaussian form

$$F_T[\phi] = \frac{1}{Z_\phi^T} \exp\left( -\frac{1}{2} \sum_\ell \frac{\Delta k}{2\pi} \sum_m \frac{\Delta k}{2\pi} \phi_\ell (\mathcal{M}_{\ell m}^T)^{-1} \phi_m \right), \tag{41}$$

with the thermal covariance given by

$$\mathcal{M}_{\ell m}^T = (1 + 2 n_{\text{BE}}(\omega_\ell)) \bar{\mathcal{M}}_{\ell m}, \tag{42}$$

wherein $n_{\text{BE}}(\omega_\ell) = 1/(e^{\beta \omega_\ell} - 1) \geq 0$ denotes the Bose-Einstein distribution. One should note that in the zero temperature limit $\beta \to \infty$ we have $n_{\text{BE}}(\omega_\ell) \to 0$ for all $\omega_\ell > 0$, such that we obtain

$$\lim_{T \to 0} F_T[\phi] = \bar{F}[\phi]. \tag{43}$$

## 2.4 Multidimensional Heisenberg uncertainty relation

For a finite number of modes, there exists a second-moment uncertainty relation for a scalar quantum field $\phi_\ell$ and its conjugate field $\pi_m$, which is the multi-dimensional generalization of (1). It reads [4, 59, 60]

$$\frac{\det(\mathcal{M} \cdot \mathcal{N})}{\det(\bar{\mathcal{M}} \cdot \bar{\mathcal{N}})} \geq 1, \tag{44}$$

which is a divergence-free formulation also in the field theory limit.

For a quantum field, the latter relation has to be understood in a matrix sense: the spectrum of the correlator product $\mathcal{M} \cdot \mathcal{N}$ is bounded from below by the (single) vacuum eigenvalue $1/4$ as $\bar{\mathcal{M}} \cdot \bar{\mathcal{N}} = \frac{1}{4} \mathbb{1}$ (see also Refs. [31, 35, 61, 62]). For pure coherent state projectors $\rho = |\alpha\rangle \langle\alpha|$, the relation becomes tight. Note that also squeezed coherent states minimize the relation (44), but exhibit disproportionate uncertainties in the quadratures. In both cases the state can be constructed from the vacuum by a unitary transformation, clearly leaving the uncertainty relation invariant. Furthermore, mixed correlations which typically arise in interacting quantum field theories, or for non-equilibrium states, can be incorporated in eq. (44) by using the full covariance matrix in the field theoretic phase space (see e.g. Refs. [4, 60]).

# 3 Relative entropic uncertainty relation

We continue with introducing the concept of functional relative entropy to overcome the divergence of the functional entropy and to quantify entropic uncertainty in a quantum field theory. Then, we state and discuss our relative entropic uncertainty relation and provide some examples.

## 3.1 Divergence of the functional entropy

The entropy corresponding to a functional probability density is in our case a functional entropy. By this we mean an entropy of an infinite dimensional random variable. More precisely, the underlying random variable is the quantum field itself. Therefore, an associated classical entropy is defined via a functional integral over all field configurations as

$$S[F] = -\int \mathcal{D}\phi \, F[\phi] \ln F[\phi].\tag{45}$$

For $d = 0 + 1$ spacetime dimensions this expression reduces to the differential entropy $S(f)$ of the position distribution $f(x)$ of a single mode.

For a finite number of modes $N \in \mathbb{N}$, the entropic uncertainty relation (2) can be generalized in a straight forward manner [4]

$$S[F] + S[G] \geq N(1 + \ln \pi),\tag{46}$$

Since the bound on the right hand side scales with the number of oscillator modes $N$, neither the continuum limit nor the infinite volume limit, which both require $N \to \infty$, are well-defined.

More precisely, independent of the state under consideration, the functional entropy $S[F]$ diverges in the field theory limit. A similar divergence appears typically for the vacuum energy expectation value $\bar{E} = \text{Tr}\{\bar{\rho}H\}$ of a quantum field. For the scalar theory defined in (15), it diverges according to

$$\bar{E} = \frac{1}{2}\int dp\, \omega(p)\, \delta(0) \to \infty.\tag{47}$$

This shows that a physically reasonable notion of energy can only be formulated as a difference to the vacuum energy, which remains finite for finitely many excitations. The choice of the vacuum $\bar{\rho}$ as a reference state appears to be natural, as it uniquely minimizes the Hamiltonian.

The divergence of the functional entropy is of similar origin. We explicitly compute the functional entropy of the vacuum functional distribution (33) in the field theory limit, which formally yields

$$S[\bar{F}] = \ln \bar{Z}_\phi + \frac{1}{2}\text{Tr}\left\{\bar{\mathcal{M}}^{-1}\bar{\mathcal{M}}\right\} = \ln \bar{Z}_\phi + \frac{1}{2}\int dp\, \delta(0) \to \infty.\tag{48}$$

As for the vacuum energy $\bar{E}$, the vacuum functional entropy $S[\bar{F}]$ is divergent as infinitely many equal and finite contributions are added up. However, in contrast to the vacuum energy, the set of states minimizing the functional entropy also comprises coherent states (for convenience we only consider states symmetric in the quadratures), which follows from the fact that the functional entropy is invariant under a displacement of field expectation values.

## 3.2 Functional relative entropy

In analogy to the energy, questions about entropic uncertainty should be asked *with respect to* suitable coherent states. This leads to the notion of a *functional relative entropy* as a measure of entropic uncertainty. We define the functional relative entropy between $F[\phi]$ and some reference distribution $\tilde{F}[\phi]$ in complete analogy to the finite dimensional case

$$S[F\|\tilde{F}] = \int \mathcal{D}\phi F[\phi]\left(\ln F[\phi] - \ln \tilde{F}[\phi]\right).\tag{49}$$

It is a non-negative quantity being zero if and only if the two distributions agree and has to be set to $+\infty$, if the support condition $\text{supp}(F) \subseteq \text{supp}(\tilde{F})$ is violated. Furthermore, it is not

symmetric and does not obey a triangle inequality, such that it should not be considered a true distance measure, but rather a divergence.

In the same way the discrete relative entropy $S(p\|\tilde{p})$ in Eq. (3) is extended to the differential relative entropy $S(f\|\tilde{f})$ in Eq. (4), one can think of the functional relative entropy $S[F\|\tilde{F}]$ as the consequent generalization of the differential one. Note again that in the limit $p(x) \to f(x)\mathrm{d}x$ the Shannon entropy $S(p)$ does not converge to the differential entropy $S(f)$ but instead tends to infinity [50], which is similar to the problems we encountered when we considered the field theory limit. In contrast, the relative entropy can always be considered a measure of distinguishability of two distributions. As a consequence, all of its properties are preserved under both continuum limits. In this sense, Eq. (49) suggests that relative entropy can be directly defined in the continuum field theory without the need for introducing an ultraviolet regulator. However, we leave a more detailed investigation in the context of algebraic quantum field theory for future work.

As for the differential relative entropy, the support condition $\mathrm{supp}(F) \subseteq \mathrm{supp}(\tilde{F})$ is always fulfilled by Gaussian reference distributions. Therefore, the functional relative entropy with respect to Gaussian reference distributions remains finite. This motivates the formulation of an entropic uncertainty relation for quantum fields in terms of functional relative entropies.

### 3.3 Deriving the relative entropic uncertainty relation

Let us start from the entropic uncertainty relation (46). To reformulate it as a relative entropic uncertainty relation, it is convenient to consider reference distributions $\tilde{F}[\phi]$ that maximize the functional entropy $S[\tilde{F}]$ under some given conditions [51]. For the particular application to a quantum field theory, we propose to employ coherent distributions as references $\tilde{F}[\phi] = F_\alpha[\phi]$, as they are known to minimize the entropic uncertainty relation (46). Then, the reference distributions are of the Gaussian form (35) and maximize the functional entropy $S[\tilde{F}]$ for a fixed covariance matrix $\bar{\mathcal{M}}$ and a given field expectation value $\phi_\ell^\alpha$.

As a consequence, the functional relative entropy of any distribution $F[\phi]$, with covariance matrix $\mathcal{M}$, and field expectation value $\varphi_\ell$ with respect to a coherent distribution $F_\alpha[\phi]$, decomposes linearly into differences of entropies and constrained quantities [51],

$$
\begin{aligned}
S[F\|F_\alpha] &= -S[F] + \ln \bar{Z}_\phi + \frac{1}{2}\mathrm{Tr}\left\{\bar{\mathcal{M}}^{-1}\mathcal{M}\right\} \\
&= -S[F] + S[\bar{F}] + \frac{1}{2}\mathrm{Tr}\left\{\bar{\mathcal{M}}^{-1}\left(\mathcal{M} - \bar{\mathcal{M}}\right)\right\} + \frac{1}{2}s\bar{\mathcal{M}}^{-1}s.
\end{aligned}
\tag{50}
$$

Therein, we used the first line of (48) together with $S[F_\alpha] = S[\bar{F}]$ and defined the difference of the field expectation values as $s_\ell \equiv \varphi_\ell - \phi_\ell^\alpha$.

Such a decomposition appears whenever the reference distribution is a maximum entropy distribution under a particular constraint. If the constrained quantities of the actual and reference distributions agree, a relative entropy equals a difference of entropies. If not, terms of differences are added, which increase the distinguishability of the actual distribution with respect to the reference distribution [51].

In this sense, the above functional relative entropy measures deviations from minimum uncertainty distributions in terms of covariance matrices and field expectation values. For a given distribution $F[\phi]$ with field expectation value $\varphi_\ell$, we can always choose a unique coherent reference distribution $F_\alpha[\phi]$ with the same field expectation value, such that $s_\ell = \varphi_\ell - \phi_\ell^\alpha = 0$. We refer to this particular coherent distribution as the *optimal coherent reference distribution*, as the functional relative entropy $S[F\|F_\alpha]$ is minimized with respect to $s_\ell$.

Following up on these considerations, we reformulate (46) solely in terms of differences of functional entropies

$$
S[F] - S[\bar{F}] + S[G] - S[\bar{G}] \geq 0.
\tag{51}
$$

Now, plugging in (50) for both entropy differences and optimal coherent reference distributions yields our main result, the relative entropic uncertainty relation (REUR)

$$S[F\|F_\alpha] + S[G\|G_\alpha] \leq \frac{1}{2}\mathrm{Tr}\left\{\bar{\mathcal{M}}^{-1}(\mathcal{M}-\bar{\mathcal{M}}) + \bar{\mathcal{N}}^{-1}(\mathcal{N}-\bar{\mathcal{N}})\right\}. \tag{52}$$

Let us emphasize that the functional probabilities $F[\phi]$ and $G[\pi]$ correspond to some arbitrary quantum field theoretic density operator $\rho$, while $F_\alpha[\phi]$ and $G_\alpha[\pi]$ are Gaussian, corresponding to a coherent state $|\alpha\rangle$. We have assumed that this coherent state has the same expectation values of fields and conjugate momenta as the state $\rho$, otherwise additional terms like the last term on the right hand side of eq. (50) would appear in (52).

### 3.4 Discussion of the relative entropic uncertainty relation

Let us make a few remarks regarding the relative entropic uncertainty relation (52) and its interpretation.

**Bound is independent of number of modes.** The resulting bound in Eq. (52) does only involve *differences* of the actual and the vacuum covariance matrices. As a consequence, the bound does not depend explicitly on the number of oscillator modes. Thus, we claim that the relative entropic uncertainty relation (52) accurately describes entropic uncertainty for all kinds of variables and degrees of freedom. In particular, it makes a non-trivial statement about entropic uncertainty for quantum fields.

**Sum of relative entropies is bounded from above.** In contrast to entropic uncertainty relations, the sum of relative entropies has an upper bound instead of a lower bound, which is similar to the relation we reported in [44]. This is due to the fact that we have chosen reference distributions which correspond to maximum entropy distributions. More precisely, the right hand side of (52) encodes the additional knowledge available about the actual distribution in terms of $\mathcal{M}$ with respect to the maximum missing information encoded in the optimal coherent reference distribution, which is completely determined by the vacuum covariance $\bar{\mathcal{M}}$.

If the state of interest is a (squeezed) coherent state itself, the relation becomes tight. In fact, if one finds that the two-point correlators $\mathcal{M}$ and $\mathcal{N}$ of an arbitrary state agree with the vacuum ones, one can conclude from (52) that this state must be coherent, while for squeezed coherent states the two sides of (52) agree but remain finite. In any other case, we get a non-trivial bound on the distinguishability of the actual distributions $F[\phi]$ and $G[\pi]$ with respect to the optimal coherent ones. Therefore, despite the fact that the bound carries some state-dependence, the relation (52) is equally tight as (46).

**Bound is of quantum origin.** To understand in which sense the uncertainty principle is encoded in (52), let us consider the classical limit for the bounds. For illustration purposes, we start with (2), for which we have to assume a finite number of modes again. To analyze the classical limit, we restore $\hbar$, such that we may take the limit $\hbar \to 0$ afterwards. We obtain[2]

$$S[F] + S[G] \geq \frac{1}{2}\ln\det\left(\hbar\bar{\mathcal{M}}\cdot\hbar\bar{\mathcal{N}}\right) \to -\infty, \tag{53}$$

in the classical limit $\hbar \to 0$, which is in accordance with the fact that classically both distributions can be arbitrarily localized simultaneously.

---

[2]We refer to [5,6] for the subtleties regarding an $\hbar$ inside a logarithm.

For the bound in our relative entropic uncertainty relation (52) we can have an arbitrary number of modes. With units restored we find

$$S[F\|F_\alpha] + S[G\|G_\alpha] \le \frac{1}{2}\mathrm{Tr}\left\{\frac{\bar{\mathcal{M}}^{-1}}{\hbar}\left(\mathcal{M} - \hbar\bar{\mathcal{M}}\right) + \frac{\bar{\mathcal{N}}^{-1}}{\hbar}\left(\mathcal{N} - \hbar\bar{\mathcal{N}}\right)\right\} \to +\infty\,, \qquad (54)$$

showing that the bound diverges to $+\infty$ in the classical limit $\hbar \to 0$. This means that classically one can distinguish some distributions from optimal coherent distributions arbitrarily well. In this sense, the relation (52) emphasizes the relevance of coherent states for the notion of uncertainty and provides a bound that can be calculated even if the relative entropies can not. Hence, we can conclude that the bound is of quantum origin and can be interpreted as the maximum (joint) distinguishability with respect to minimum uncertainty states allowed by the uncertainty principle.

## 3.5 Examples

To exemplify the independence of the bound in (52) on the number of modes $N$, we consider excited states and the thermal state for discrete as well as continuous degrees of freedom.

### 3.5.1 Excited states

We start with a state that carries finitely many excitations corresponding to free (quasi-)particles, introduced in subsection 2.3. For example, exciting a single momentum mode in a relativistic quantum field theory generates a freely moving, completely delocalized particle with definite momentum. For a finite number of excitations, the relation (52) makes a non-trivial statement about entropic uncertainty, which is what we will show in the following.

As excited states have vanishing field expectation values, the vacuum serves as an optimal reference distribution in the relative entropic uncertainty relation (52). In order to determine the bound, we have to compute the covariance matrix, which follows from the corresponding functional distribution (38). We find the relation

$$\mathcal{M}_{\ell m} = \int \mathcal{D}\phi \left[\phi_\ell \phi_m \prod_{k\in\mathfrak{I}} \frac{1}{n_k!} H_{n_k}^2\left(\frac{\phi_k}{\sqrt{\bar{\mathcal{M}}_{kk}}}\right)\right]\bar{F}[\phi]\,. \qquad (55)$$

The latter expression contains Gaussian integrals over all modes. To solve them, we make a distinction of cases. First, using the fact that Gaussian integrals of odd polynomials evaluate to zero, we immediately obtain that $\mathcal{M}$ is – just as its vacuum counterpart $\bar{\mathcal{M}}$ – a diagonal matrix (possibly in a continuous sense). Furthermore, if $\phi_\ell$ is a non-excited mode, i.e., if $\ell \notin \mathfrak{I}$, the corresponding variance $\mathcal{M}_{\ell\ell}$ is equal to the vacuum variance $\bar{\mathcal{M}}_{\ell\ell}$.

This leaves us with the last possibility, namely the case where $\phi_\ell$ is an excited mode for which we have to calculate the variance $\mathcal{M}_{\ell\ell}$. We use the orthogonality relation between the probabilist's Hermite polynomials defined in (39),

$$\int_{-\infty}^{+\infty} \mathrm{d}\phi_\ell \, H_a(\phi_\ell) H_b(\phi_\ell) e^{-\frac{1}{2}\phi_\ell^2} = \sqrt{2\pi}\, a!\, \delta_{ab}\,, \qquad (56)$$

together with their recurrence relation

$$H_{a+1}(\phi_\ell) = H_1(\phi_l) H_a(\phi_\ell) - a H_{a-1}(\phi_\ell)\,, \qquad (57)$$

which leads to

$$\mathcal{M}_{\ell\ell} = \bar{\mathcal{M}}_{\ell\ell}(1 + 2n_\ell)\,, \qquad (58)$$

for $\ell \in \mathfrak{I}$.

Combining all the above considerations, the covariance matrix of a general excited state can be written as

$$\mathcal{M}_{\ell m} = \bar{\mathcal{M}}_{\ell m} + \sum_{k \in \mathfrak{I}} \frac{2 n_k}{\bar{\mathcal{M}}_{kk}} \bar{\mathcal{M}}_{\ell k} \bar{\mathcal{M}}_{mk} . \tag{59}$$

This result has a very clear interpretation. The diagonal components of the vacuum covariance acquire an additive term accounting for the $n_k$ excitations in the excited mode $k$. Components corresponding to unexcited modes remain unmodified.

Then, we obtain for one of the terms in the bound of the relative entropic uncertainty relation (52)

$$\begin{aligned}
\frac{1}{2} \mathrm{Tr} \left\{ \bar{\mathcal{M}}^{-1} (\mathcal{M} - \bar{\mathcal{M}}) \right\} &= \sum_{\ell} \frac{\Delta k}{2\pi} \sum_{m} \frac{\Delta k}{2\pi} \bar{\mathcal{M}}_{\ell m}^{-1} \sum_{k \in \mathfrak{I}} \frac{n_k}{\bar{\mathcal{M}}_{kk}} \bar{\mathcal{M}}_{\ell k} \bar{\mathcal{M}}_{mk} \\
&= \sum_{k \in \mathfrak{I}} \frac{n_k}{\bar{\mathcal{M}}_{kk}} \bar{\mathcal{M}}_{kk} \\
&= \sum_{k \in \mathfrak{I}} n_k ,
\end{aligned} \tag{60}$$

where we used (29). In the field theory limit, the expression above is still well-behaved as all divergencies cancel out. Employing (31), we obtain

$$\begin{aligned}
\frac{1}{2} \mathrm{Tr} \left\{ \bar{\mathcal{M}}^{-1} (\mathcal{M} - \bar{\mathcal{M}}) \right\} &= \int \frac{\mathrm{d}p}{2\pi} \frac{\mathrm{d}q}{2\pi} \bar{\mathcal{M}}^{-1}(p,q) \sum_{k \in \mathfrak{I}} \frac{n_k}{\bar{\mathcal{M}}(k,k)} \bar{\mathcal{M}}(p,k) \bar{\mathcal{M}}(q,k) \\
&= \sum_{k \in \mathfrak{I}} \frac{n_k}{\bar{\mathcal{M}}(k,k)} \bar{\mathcal{M}}(k,k) \\
&= \sum_{k \in \mathfrak{I}} n_k ,
\end{aligned} \tag{61}$$

showing that the bound remains unmodified.

In order to render all quantities finite and well-defined during the calculation of the bound of the relative entropic uncertainty relation, one can use suitably chosen wave packet states, as demonstrated in subsection 3.7 for the free one-particle state. As it is not essential for our investigation, we abstain from a detailed discussion of this issue for the case of the more general excitations.

After performing an analogous calculation for the covariance matrix $\mathcal{N}$, the bound of the relative entropic uncertainty relation (52) evaluates to

$$\frac{1}{2} \mathrm{Tr} \left\{ \bar{\mathcal{M}}^{-1} (\mathcal{M} - \bar{\mathcal{M}}) + \bar{\mathcal{N}}^{-1} (\mathcal{N} - \bar{\mathcal{N}}) \right\} = \sum_{k \in \mathfrak{I}} 2 n_k . \tag{62}$$

As anticipated, the bound does not scale with the number of oscillator modes $N$, but linearly with the total number of excitations. Thus, we have shown that the right hand side of (52) is finite.

Let us now consider the left hand side of (52). We omit the calculation of the functional relative entropies for general excitations, which is a highly non-trivial task[3]. As a simple and instructive example, let us consider a single excitation in the mode $k$, which corresponds to a freely moving particle with energy $\omega(k)$ in the field theory limit. After a straightforward exercise in Gaussian integration one finds

$$S[F_1 \| \bar{F}] + S[G_1 \| \bar{G}] = 4 - \ln 4 - 2\gamma , \tag{63}$$

---

[3]See Refs. [63–65] for the differential entropy of number eigenstates in various dimensions. Note that in contrast the Wehrl entropy, which is the entropy associated with the Husimi $Q$-distribution defined in (25), can be calculated easily for arbitrary excitations [20].

where $\gamma \approx 0.577$ is the Euler-Mascheroni constant. The uncertainty deficit, i.e., the difference between the right hand side and the left hand side of the REUR (52) is in agreement with the one-dimensional result [63]. Therefore, we find that the entropic uncertainties of a free particle in a quantum field theory and a single excited oscillator mode attain the same numerical values. The latter result is expected to generalize to arbitrary excitations.

### 3.5.2   Thermal state

As a second example we consider the thermal state, where again the vacuum acts as an optimal reference distribution. With the thermal covariance matrix at hand (cf. eq. (42)) we can calculate the bound of the relative entropic uncertainty relation (52), which yields in the discrete case

$$\frac{1}{2}\text{Tr}\left\{\bar{\mathcal{M}}^{-1}(\mathcal{M}^T - \bar{\mathcal{M}})\right\} = L\sum_\ell \frac{\Delta k}{2\pi}n_{\text{BE}}(\omega_\ell). \tag{64}$$

In the continuum limit, the bound remains finite due to the exponential fall-off of the Bose-Einstein distribution $n_{\text{BE}}(\omega_\ell)$. However, the bound is also proportional to the length $L$ of the interval under consideration. Therefore, in the infinite volume limit we obtain

$$\frac{1}{2}\text{Tr}\left\{\bar{\mathcal{M}}^{-1}(\mathcal{M}^T - \bar{\mathcal{M}})\right\} = 2\pi\delta(0)\int\frac{\text{d}p}{2\pi}n_{\text{BE}}(\omega(p)), \tag{65}$$

where $L = \frac{2\pi}{\Delta k} \to 2\pi\delta(0)$ represents an infinite volume factor. Since the integral remains finite for non-negative inverse temperatures and masses $\beta, m > 0$, the bound is still well-defined up to this infinite volume factor.

Such infinite volume factors appear generically for thermal states in a quantum field theory, in particular also for the energy difference with respect to the vacuum $E_T - \bar{E} \sim \delta(0)$. This problem is typically circumvented by considering energy *densities* instead of absolute energies, or by treating the quantum field in a finite volume. A similar line of reasoning can be used to obtain functional relative entropy densities, leading to a divergence-free notion of entropic uncertainty.

Since the thermal state is of Gaussian form, one can easily calculate the left hand side of the relative entropic uncertainty relation (52), which gives for a discrete set of modes

$$S[F_T\|\bar{F}] + S[G_T\|\bar{G}] = L\sum_\ell \frac{\Delta k}{2\pi}\left[2n_{\text{BE}}(\omega_\ell) - \ln(1 + 2n_{\text{BE}}(\omega_\ell))\right]. \tag{66}$$

To exemplify the behavior of the relative entropic uncertainty relation in this case, we consider one mode, i.e. $N = L = \epsilon = 1$ and $\Delta k = 2\pi$, with a given frequency $\omega$ (cf. Figure 1). The uncertainty deficit is non-negative and approaches zero in the zero temperature limit $\beta \to \infty$ as $n_{\text{BE}}(\omega_\ell) \to 0$.

### 3.6   Relation to the Heisenberg relation

For a single mode, the relation by Białynicki-Birula and Mycielski (2) is stronger than and therefore implies the Heisenberg uncertainty relation (1) (see e.g. [4, 66]). Thus, we ask the question how the relative entropic uncertainty relation (52) is related to the second-moment uncertainty relation (44).

We begin with reformulating the left hand side of relative entropic uncertainty relation (52). Instead of choosing the optimal coherent functional densities as reference distributions for the relative entropies, we instead consider Gaussian reference distributions $\tilde{F}[\phi]$ and $\tilde{G}[\pi]$ which have the same expectation values and two-point correlation functions as the actual functional densities, i.e., $\tilde{\mathcal{M}} = \mathcal{M}$ and $\tilde{\varphi}_\ell = \varphi_\ell$ (analogously for the momentum field $\pi$).

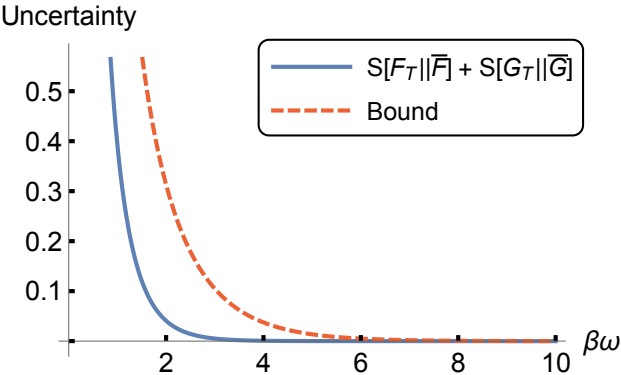

Figure 1: Both sides of (66) are shown as a function of $\beta$ for one mode with fixed frequency $\omega$. For large $\beta$, the thermal state $\rho_T$ approaches the ground state $\bar{\rho}$ and the bound becomes tight. For small $\beta$, the state $\rho_T$ is highly mixed and therefore the uncertainty grows. For $\beta \to 0$ the state approaches the uniform distribution in which case the entropic uncertainty is unbounded.

These reference distributions can be considered the *optimal* Gaussian reference distributions measured in terms of the relative entropies (cf. subsection 3.3). Using (50) and (51), the relative entropic uncertainty relation (52) can be rewritten as

$$S[F\|\tilde{F}] + S[G\|\tilde{G}] \leq \Delta S[\tilde{F}] + \Delta S[\tilde{G}], \tag{67}$$

where

$$\Delta S[\tilde{F}] = S[\tilde{F}] - S[\bar{F}] = \frac{1}{2} \ln \frac{\det \mathcal{M}}{\det \bar{\mathcal{M}}}, \tag{68}$$

is the functional entropy difference between the optimal Gaussian reference distribution and the vacuum (or any coherent state). The right hand side of this uncertainty relation can be calculated explicitly

$$\Delta S[\tilde{F}] + \Delta S[\tilde{G}] = \frac{1}{2} \ln \frac{\det(\mathcal{M} \cdot \mathcal{N})}{\det(\bar{\mathcal{M}} \cdot \bar{\mathcal{N}})}. \tag{69}$$

Then, we can reformulate the uncertainty relation as

$$\frac{\det(\mathcal{M} \cdot \mathcal{N})}{\det(\bar{\mathcal{M}} \cdot \bar{\mathcal{N}})} \geq e^{2(S[F\|\tilde{F}] + S[G\|\tilde{G}])} \geq 1. \tag{70}$$

Hence, the relative entropic uncertainty relation (52) is stronger and implies the relation (44) in the same sense as in the one-dimensional case. Moreover, the latter chain of inequalities shows that deviations of the distributions from Gaussianity increase the uncertainty product.

## 3.7 Averaged fields and measurability

At last, let us discuss the measurability of the fields $\Phi$ and $\Pi$ and their corresponding functional probability densities $F[\phi]$ and $G[\pi]$. We note that a quantum field taken at a single point in space or at a definite momentum is not a proper observable. Mathematically, the object $\Phi(x)$ or $\Phi(p)$, respectively, is an operator-valued distribution rather than an operator. Physically, a field at a single point can never be resolved with arbitrary precision by any measurement device as this would require infinite energy. As a consequence, the functional probability densities $F[\phi]$ and $G[\pi]$ also do not constitute proper observables.

In order to render the field an observable, we have to average $\Phi(p)$ with a test function [67, 68]. A convenient choice for a class of test functions are the Schwartz functions $\mathcal{A}(p) \in \mathscr{S}(\mathbb{R})$.

These include, for example, normalized Gaussian functions. By choosing a narrow Gaussian with expectation value $\mu_p = k$, denoted by $\mathcal{A}_k(p)$, we can approximate an excitation in a single momentum mode $k$. Then, the averaged field operator is defined as

$$\Phi(\mathcal{A}_k) \equiv \int \frac{dp}{2\pi} \, \mathcal{A}_k(p) \, \Phi(p) \, . \tag{71}$$

For a discussion of the Heisenberg uncertainty relation for similarly defined averaged fields, see [69].

To illustrate the behavior of the relative entropic uncertainty relation (52) for averaged fields, we define a *wave packet one-particle state* as

$$\Psi_1[\phi] = \frac{\phi[\mathcal{A}_k]}{\sqrt{\bar{\mathcal{M}}(k,k)}} \bar{\Psi}[\phi] \, , \tag{72}$$

with $\phi[\mathcal{A}_k]$ the eigenvalue of the averaged field operator $\Phi(\mathcal{A}_k)$ defined in (71). The probability density of this state reads

$$F_1[\phi] = \frac{\phi^2[\mathcal{A}_k]}{\bar{\mathcal{M}}(k,k)} \bar{F}[\phi] \, , \tag{73}$$

and is now a proper, measurable functional of the field. The corresponding normalization constant $Z_\phi^1$ is still proportional to the vacuum normalization $\bar{Z}_\phi$,

$$Z_\phi^1 = \bar{Z}_\phi \, \bar{\mathcal{M}}(k,k) \, , \tag{74}$$

where the proportionality constant

$$\bar{\mathcal{M}}(k,k) = \pi \int \frac{dp}{2\pi} \frac{\mathcal{A}_k^2(p)}{\omega(p)} \, , \tag{75}$$

is now finite.

On the level of the relative entropic uncertainty relation, considering the wave packet one-particle state renders *all* involved quantities finite. This can be seen best by writing out the bound

$$\frac{1}{2} \text{Tr} \left\{ \bar{\mathcal{M}}^{-1} \left( \mathcal{M}^1 - \bar{\mathcal{M}} \right) \right\} = \frac{\pi}{\bar{\mathcal{M}}(k,k)} \int \frac{dp}{2\pi} \frac{\mathcal{A}_k^2(p)}{\omega(p)} = 1 \, , \tag{76}$$

where in the second step two finite instead of two infinite quantities canceled out.

Therefore, the relative entropic uncertainty relation accurately describes entropic uncertainty also for averaged fields. Specifically, the bound once again remains unmodified, exemplifying its generality.

# 4 Conclusion and Outlook

In summary, we have presented a relative entropic uncertainty relation describing entropic uncertainty between a scalar field and its conjugate momentum field with respect to optimal coherent states. Motivated by the pathological behavior of the Shannon entropy in the continuum limit, and the divergence of the vacuum energy in a quantum field theory, we suggested the use of functional *relative* entropies to quantify entropic uncertainty.

Due to the independence of the resulting relative entropic uncertainty relation from the number of modes, we obtained a well-defined and divergence-free entropic uncertainty relation

in the field theory limit. This was shown exemplary by considering arbitrary excitations and the thermal state. While in the former case we obtained finite results for the continuum as well as the infinite volume limit, in the latter case we argued to consider either finite volumes or relative entropy densities.

For future theoretical work it is particularly interesting to formulate other known entropic uncertainty relations in a field theory sense, for example the relation by Frank and Lieb [29] or the Wehrl-Lieb inequality [70, 71]. Furthermore, one may extend the relative entropic uncertainty relation to include (quantum) memory (cf. Refs. [24, 72]).

As entropic uncertainty relations play a crucial role in the context of quantum entanglement, we propose to employ our relative entropic uncertainty relation to constrain entanglement in quantum field theories. In this way, one may be able to obtain criteria being capable of certifying entanglement between spacetime regions.

Another direction for further research is to study other field theories, which can be divided into two major directions. First, one may consider field theories of a different type, e.g. fermionic degrees of freedom or gauge fields. While the former may be interesting especially for experimental applications, the latter introduces the challenge that not all field configurations contribute independently to the functional integral. Second, one may study interacting theories. We expect that the main results of this work can be extended at least into the perturbative regime.

Furthermore, one may investigate the functional relative entropy in the context of algebraic quantum field theory to see whether one can find a generic formula in terms of expectation values of relative modular operators.

Lastly, it would be of great interest to study the relative entropic uncertainty relation in experimental setups. For example, one could investigate Bose-Einstein condensates and estimate the bound on entropic uncertainty of a freely travelling phonon.

## Acknowledgements

The authors thank Martin Gärttner for fruitful discussions and careful reading of the manuscript, Alexander Vikman for useful correspondence and the two anonymous referees for their criticism, which helped to improve the presentation of the manuscript. This work is supported by the Deutsche Forschungsgemeinschaft (DFG, German Research Foundation) under Germany's Excellence Strategy EXC 2181/1 - 390900948 (the Heidelberg STRUCTURES Excellence Cluster) and under 273811115 – SFB 1225 ISOQUANT as well as FL 736/3-1.

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
