# Peer review of "Relative entropic uncertainty relation for scalar quantum fields"

_SciPost Physics, doi:SciPost Phys. 12, 089 (2022)_

## Round 2 · Referee Report · Anonymous (Referee 1) · 2021-11-23

Strengths

1 - The computations are not particularly difficult and easy to reproduce.
2 - Conciseness.

Weaknesses

1 - Not really an original work. Most of the results appear just as a multidimensional generalization of a single quantum harmonic oscillator. 2- The conclusions are not really emphasized, and one has to go through all the manuscript to understand what is the point of the whole discussion.

Report

In this paper, the authors study the relative entropic uncertainty relation for scalar quantum fields. They find that relative measures of uncertaintes, comparing the state under analysis and a reference state, are well-behaved even in the continuum limit, meaning that they could be good notions for states of quantum field theories.
The work points out interesting observations in the context of quantum information and its relation with quantum field theory. While I do have some comments and few suggestions to improve the presentation, these are minor and I recommend publication.
Here is the list of comments and suggestion:

Requested changes

1- Page 5, after Eq. 39: "...with the thermal covariance being proportional to the vacuum one". This sentence does not really make sense; the two matrices are not proportional, since the "thermal" factor (1+2n_{BE}(\omega_l)) depends on the index. Maybe "...with the thermal covariance given by..." would be better. 2- Page 5, just before Eq. 42. "For a free theory in an equilibrium state, the mixed correlations... ... vanish". If I am right for "equilibrium state" the author means "invariant under time reversal", which would imply the relation < \phi \pi +\pi \phi> =0. The sentence is probably too concise to be understood at first glance and at least a reference for that is needed. 3 - Page 5, just after Eq. 42. "the eigenvalues of the correlator product MN are at least the eigenvalues...". I do not really understand this sentence. In the ground state \bar{M}\bar{N} =1/4, so the only eigenvalue is 1/4; what does it means that the eigenvalues of MN are also eigenvalues of \bar{M}\bar{N} ? 4 - The conclusions of the work are hidden. The Eq. 50 is considered as the main result of the work, and it would be better to anticipate it in the introduction or emphasize it again in the conclusions. 5- It is not particularly clear, at least at first reading, what is the regime of validity of the results. While the Heisenberg relation or the "Bialynicki-Birula and Myciel ski" formula are generic, the focus of the manuscript appears to be restricted to certain class of states obtained as eigenstates or thermal states of the Klein Gordon theory. An improvement of the presentation would be the insertion of a precise statement (in the introduction) regarding the range of validity of certain inequalities. 6 - I would also suggest emphasizing more the connection with a previous work of the authors regarding the same topic (Relative entropic uncertainty relation, ref. [31]). Probably even a small comment/appendix summarizing the results available for a single random variable would be helpful.

---

## Round 2 · Referee Report · Anonymous (Referee 2) · 2021-12-3

Strengths

  1. The problem is very well motivated in the introduction. Quickly grab the reader's attention.
  2. The computations are very clear and can be easily reproduced/verified.
  3. The result is based on precise mathematical calculations, and does not involve "leap of faith" arguments.
  4. It is very well written.

Weaknesses

In addition to the major drawback that I point out in the section report, I would like to point out the following:

  1. The result is "hidden" in the text. It forces the reader to go through a lot of technical details to get to it. It would be much better if the authors anticipated or referenced the result at the end of the introduction.
  2. SUGGESTION: The authors justify the use of relative entropy to construct the uncertainty principle based primarily on the fact that relative entropy is finite at the limit of the continuum. But the same happens for difference of two von Neumann entropies. I think the authors could give better reasons for that choice. One of them is that the relative entropy can be defined and calculated (in principle) directly in the continuum QFT, without the need of introducing a cutoff (Araki formula).

Report

In the present work the authors address the original problem of finding an entropic uncertainty principle for a scalar quantum field. Entropic certainty relations are a very important subject of study in modern physics because they have a very wide spectrum of applications: from quantum foundations to experiments. Contrary to what is often the case with finite quantum systems, where the von Neumann (vN) entropy is often used to formulate uncertainty relations, the authors uses the relative entropy. As it was pointed out for the authors, the main reason is that the relative entropy (for a correct choice of states) has a finite continuum limit, whereas the vN entropy does not. This allows them to report a result that holds even in the continuum QFT.

The main result of the work is the relative entropy uncertainty relation shown in equation (50). From my understanding on the subject, this expression is not on the same level as other known uncertainty relations because the r.h.s. still depends on the underlying states. What I expect for a uncertainty relation (like the one in eqs. (1) or (2)) is an inequality that indicates that the the sum of two entropic measures (or the product of two standard deviations) is greater or less than a bound which may depend on the theory but not on the state(s). I think the authors have to better argue in which sense their result can still be considered as an uncertainty relation at the same level as the other uncertainty relations that one can find in the literature. Or in which way one has to understand inequality (50) to affirm that the bound in the r.h.s. is still relevant and useful. Typically, the usefulness of an uncertainty relation is that even when one is not able to calculate the l.h.s, one can easily have a bound by just reading the r.h.s. This is not the case of the relation presented here.
Furthermore, according to eq. (66), it seems that eq. (50) can be understood as an improvement of the Robertson-Schrödinger uncertainty relation for the quantity that appears in the l.h.s. of (66). This would be another way of looking at the same result, but it still implies an uncertainty relation whose “bound” is state dependent.
I would consider this like a major revision. If the authors could address this problem, I would recommend this work for publication.

Requested changes

Some of these are requested changes, others are minor questions for the authors:

  1. The sum in equation (3) would run from 1 to N to be compatible with the boundary condition \phi_0 = \phi_n.
  2. In what sense eq. (5) is a unitary transformation? If I understood correctly, up this point the computation is in classical field theory. The quantization comes later.
  3. In eq. (6), should it say \tilde{\phi} and \tilde{\pi}?
  4. In eq. (8), the fields are written in capital letters contrary to what happened before. Presumably this change was made for the purpose of differentiating classical from quantum variables. The authors should stress that in the text.
  5. For eqs. (14) and (15), it should be emphasized that \phi_l and \pi_l are real numbers, and that the basis are orthonormal.
  6. I would like to ask the authors if they can provide a reference for equations (36-37). Otherwise, if it is a relation found by them, I would like to ask them about the relevant points in the calculation of such a relation.
  7. From (50), can it be inferred that if a state has the same two-point functions as a coherent state, then it must be coherent?
  8. It is not entirely clear to me what the authors are saying in the first paragraph of the 2nd column on page 9 (the one that starts with: “We begin by reformulating ...”). If one chooses a state that has the same two-point correlators as the reference state, then the the r.h.s. of (50) is identically zero. Is that correct?

---

## Round 3 · Referee Report · Anonymous (Referee 1) · 2021-12-21

Report

We really thank the authors for the detailed answers they gave to each comment we pointed out. The current version of the work is now clearer and we recommend its publication for this Journal

---

## Round 3 · Referee Report · Anonymous (Referee 2) · 2021-12-23

Strengths

Adding to the strengths that I pointed out in the first submission, I would like to add:

  1. The anticipation of the the result in the introduction, with an adequate explanation and the appropriate reference to the equation in the main text below.

  2. The explanation in the introduction about how the relative entropy is a good measure of uncertainty, and why it is convenient to use it in QFT.

Weaknesses

The weaknesses that I pointed out in the first submission of the manuscript were correctly addressed by the authors, where they made the corresponding changes.

Report

I would like to thank the authors for clarifying my concerns and improving the manuscript. Now I agree with the authors that regardless the bound coming from the REUR is state dependent, it still describes the degree of uncertainty of the underlying state. In these lines, I appreciate that the authors strength the quantum origin of this bound as the did in section D. I recommend that this version of the manuscript be published in the journal.

Requested changes

No further changes are requested.

---

## Round 3 · Author Response

We thank the referees for carefully reading our manuscript and their very constructive reports.

In the following, we will address the critique of the second referee. We respond to every item in 'Requested changes' of both reports below 'List of changes'.

  • Concerning 'result is hidden in the text':

As this was also pointed out by the first referee, we added a two paragraphs in the introduction anticipating the main results and their ranges of validity.

  • Concerning 'motivating the use of relative entropy':

To motivate better the use of relative entropy, we devoted a new paragraph in the introduction to relative entropy. We argue that relative entropy is more universal than entropy in the sense that it its properties are the same in many occasions. Also, we reformulated the paragraph below eq. (49) and added a sentence that the functional relative entropy may possibly be defined directly in the continuum theory. Additionally, we have divided subsection III. A. into A. and B. to prevent the appearance of an overfull subsection.

However, we think that our functional relative entropy does not coincide with the quantum relative entropy, which is what follows from the Araki formula. Also, we understand the von Neumann entropy as the quantum entropy associated with the density operator. In this sense, we believe that entropic uncertainty relations are not formulated in terms of von Neumann entropies, but rather in terms of the classical entropies, i.e. Shannon entropies (for discrete variables) or differential entropies (for continuous variables).

  • Concerning 'state-dependence of the bound':

We agree that many well-known (entropic) uncertainty relations have a state-independent bound. However, there are also many bounds which are state-dependent and especially nowadays some state-dependent bounds have important applications when it comes to entanglement witnessing (cf. Ref. [5], eq. (331) therein). For example, the famous Robertson relation is formulated in terms of an expectation value in the right hand side. The state-independence of the derived Heisenberg-Kennard relation (eq. (1)) is rather a consequence of the fact that the commutation relation between position and momentum gives a c-number. If one considers spins equipped with an SU(2) algebra instead, the bound becomes explicitly state-dependent.

Also for entropic uncertainty relations state-dependent bounds are known. For example, for position and momentum an alternative bound to eq. (2) is given by ln 2 \pi + S(\rho), where S(\rho) denotes the von Neumann entropy of the quantum state \rho (cf. Ref. [29] or Ref. [5] and eq. (267) therein). For discrete variables, the Maassen-Uffink relation has also been improved by adding S(\rho) to the bound (cf. Ref. [5], eq. (47) therein). Let us point out that state-dependent bounds are often tighter. In fact, as variance as well as entropy are concave, the (entropic) uncertainty should become minimal for pure states. Hence, one may tighten a pure-state bound by adding quantities which measure the mixedness of the quantum state.

However, state-dependence of a bound can also arise from a reformulation and is not necessarily related to tightness. In particular, the state-dependence of our bound does not mean that it is tighter than the BBM relation for a single mode. It is rather a consequence of using relative entropies instead of entropies. In fact, the uncertainty deficit, i.e. the difference between the left hand side and the right hand side, agrees with the uncertainty deficit of the BBM relation.

  • Concerning 'in which sense the REUR expresses the uncertainty principle':

We thank the referee in particular for requiring are more detailed statement regarding how the relative entropic uncertainty relation expresses the uncertainty principle. We have restructured section III C. (before: section III B.) by dividing it into C. (Deriving the relative entropic uncertainty relation) and D. (Discussion of the relative entropic uncertainty relation). While the new subsection III. C. remains unchanged, we have extended D. by a new paragraph. Therein, we argue that the sum of considered relative entropies is not bounded in a classical theory, allowing us to conclude that the bound is purely of quantum origin.

  • Concerning 'relation between entropic uncertainty relation and Robertson-Schrödinger relation':

As correctly pointed out by the referee, the REUR (or equivalently the BBM relation for finitely many modes) is an improvement of the variance-based formulation in the sense of eq. (70). We would like to point out that this has been studied in Ref. [4] (around eq. (46) therein) in detail and allows for the interpretation that the bound in eq. (1) is lifted by an exponentiated sum of relative entropies showing that that the entropic uncertainty relation eq. (1) is tighter than the variance-based relation eq. (2) whenever the distributions are non-Gaussian. This again shows that state-dependence of a bound may arise through a reformulation of an uncertainty relation.

We want to thank both referees again for their constructive criticism which helped us to improve the manuscript further. For the referees' convenience, we have attached a latexdiff pdf which highlights all the changes made since submission. We believe that it is now ready for publication in SciPost Physics.

---

## Round 3 · List of Changes

Warnings issued while processing user-supplied markup:

  • Inconsistency: plain/Markdown and reStructuredText syntaxes are mixed. Markdown will be used.
    Add "#coerce:reST" or "#coerce:plain" as the first line of your text to force reStructuredText or no markup.
    You may also contact the helpdesk if the formatting is incorrect and you are unable to edit your text.

To first referee:

1- Page 5, after Eq. 39: "...with the thermal covariance being proportional to the vacuum one". This sentence does not really make sense; the two matrices are not proportional, since the "thermal" factor $(1+2n_{BE}(\omega_l))$ depends on the index. Maybe "...with the thermal covariance given by..." would be better.

Yes, we implemented the requested change.

2- Page 5, just before Eq. 42. "For a free theory in an equilibrium state, the mixed correlations... ... vanish". If I am right for "equilibrium state" the author means "invariant under time reversal", which would imply the relation $< \phi \pi +\pi \phi> =0$. The sentence is probably too concise to be understood at first glance and at least a reference for that is needed.

The requirement $< \phi \pi +\pi \phi> =0$ is actually not needed for the argument we had in mind and hence we dropped it. To be a bit more precise: In a free scalar field theory, the covariance matrix in phase space is symmetric and hence can be block-diagonalized (by rotations in phase space). For some states, this is already the case in the canonical basis given by \phi and \pi, including all examples we consider in our work. Now, there exist two second-moment based uncertainty relations (see the discussion in Ref. [59]). One is the Robertson-Schrödinger relation, expressed in terms of the phase space covariance matrix, and one is the multidimensional generalization of the Heisenberg relation eq. (1), which is a special case of the former if the fields are rotated such that the mixed correlations vanish. Otherwise, the latter is weaker than the former. As our REUR is only capable of implying the multidimensional Heisenberg relation, we changed "Robertson-Schrödinger" to "multidimensional Heisenberg" everywhere. In fact, it is an open problem to find an EUR which implies the general Robertson-Schrödinger relation even for a single oscillator (see again Ref. [59]).

3 - Page 5, just after Eq. 42. "the eigenvalues of the correlator product MN are at least the eigenvalues...". I do not really understand this sentence. In the ground state \bar{M}\bar{N} =1/4, so the only eigenvalue is 1/4; what does it means that the eigenvalues of MN are also eigenvalues of \bar{M}\bar{N}? --

The important point is that the eigenvalues of the correlator product MN are bounded from below. We have clarified this now by adjusting the corresponding sentence.

4 - The conclusions of the work are hidden. The Eq. 50 is considered as the main result of the work, and it would be better to anticipate it in the introduction or emphasize it again in the conclusions.

As this was also requested by the second referee, we devoted a new paragraph in the introduction to the main result. In particular, we emphasized the REUR itself, that it holds for oscillators as well as fields and that that the considered sum of relative entropies is non-trivially bounded from above by the uncertainty principle.

5- It is not particularly clear, at least at first reading, what is the regime of validity of the results. While the Heisenberg relation or the "Bialynicki-Birula and Mycielski" formula are generic, the focus of the manuscript appears to be restricted to certain class of states obtained as eigenstates or thermal states of the Klein Gordon theory. An improvement of the presentation would be the insertion of a precise statement (in the introduction) regarding the range of validity of certain inequalities.

We included a new paragraph at the end of the introduction clarifying that we focus on theories where the field and the conjugate momentum field fulfill a bosonic commutation relation and where the vacuum is Gaussian (i.e. free theories).

6 - I would also suggest emphasizing more the connection with a previous work of the authors regarding the same topic (Relative entropic uncertainty relation, ref. [31]). Probably even a small comment/appendix summarizing the results available for a single random variable would be helpful.

Following up on a suggestion of the second referee, we introduced the discrete and continuous relative entropies in the introduction in eqs. (3) and (4) (which is why all subsequent equation numbers are shifted by two compared to the previous version) and argued why relative entropy may be preferred over entropy. Thereupon, we summarized the main result of our previous work, i.e. that we found an entropic uncertainty relation which holds for discrete as well as continuous variables formulated in terms of relative entropy.

To second referee:

1. The sum in equation (3) would run from 1 to N to be compatible with the boundary condition $\phi_0 = \phi_n$.

Thanks, we corrected this mistake.

2. In what sense eq. (5) is a unitary transformation? If I understood correctly, up this point the computation is in classical field theory. The quantization comes later.

Both $\tilde{\phi}$ and $\tilde{\pi}$ are $N$-dimensional vectors with complex entries. Eq. (7) describes the action of an unitary $N \times N$ matrix on this vector, written in terms of their components. In this sense, this transformation is not related to quantization. It is a unitary transformation of the classical fields just as the discrete Fourier transform performed in eq. (6).

3. In eq. (6), should it say \tilde{\phi} and \tilde{\pi}?

No, in this case we would need absolute values of the tilde fields as they are complex-valued. For reasons of convenience, the goal was to write the Hamiltonian in terms of real fields. This is why we implemented the additional unitary transformation in eq. (7). Please note that the transformation in eq. (7) keeps the Hamiltonian diagonal.

4. In eq. (8), the fields are written in capital letters contrary to what happened before. Presumably this change was made for the purpose of differentiating classical from quantum variables. The authors should stress that in the text.

This choice of notation was clarified already in the "Notation" paragraph at the end of the introduction. However, we added "hermitian quantum field operators" above eq. (10) to avoid misunderstandings.

5. For eqs. (14) and (15), it should be emphasized that $\phi_l$ and $\pi_l$ are real numbers, and that the basis are orthonormal.

Following the request, we added "orthonormal" above eq. (16) and a half sentence below eq. (17).

6. I would like to ask the authors if they can provide a reference for equations (36-37). Otherwise, if it is a relation found by them, I would like to ask them about the relevant points in the calculation of such a relation.

We use the creation operators defined in eq. (36) and act with them on the vacuum wave functional in eq. (37) to obtain eq. (38) with the Hermite polynomials in eq. (39). The same strategy is used in Ref. [51]. Hence, we added this reference in the first paragraph of the subsection. One can find such calculations in chapter 10 of Ref. [51] for one excitation and in chapter 11 for multiple excitations.

7. From (50), can it be inferred that if a state has the same two-point functions as a coherent state, then it must be coherent?

This is true. Let us emphasize that we need that both M and N agree with their vacuum counterparts to conclude that the corresponding state is coherent. If only one of them is set to the vacuum expression, this is not the case anymore. In our opinion, this is worth mentioning, and consequently we added a sentence in the last paragraph of section III. D. under bullet point 'b. Sum of relative entropies is bounded from above'.

8. It is not entirely clear to me what the authors are saying in the first paragraph of the 2nd column on page 9 (the one that starts with: “We begin by reformulating ...”). If one chooses a state that has the same two-point correlators as the reference state, then the the r.h.s. of (50) is identically zero. Is that correct?

We have added a few words in this paragraph, which clarify that the REUR in eq. (52) has a different left hand side than the relation we derive afterwards. Nevertheless, it is always true, for both formulations eq. (52) and eq. (67), that the bound is zero whenever one chooses a state which has the same two-point functions as the reference state. However, the physical meaning of this choice depends on the choice of reference state, which is the crucial difference between eqs. (52) and (67).

---

## Editorial Decision

published